# Clinical Outcomes in Patients Aged 80 Years or Older Receiving Non-Invasive Respiratory Support for Hypoxemic Acute Respiratory Failure Consequent to COVID-19

**DOI:** 10.3390/jcm11051372

**Published:** 2022-03-02

**Authors:** Andrea Vianello, Nello De Vita, Lorenza Scotti, Gabriella Guarnieri, Marco Confalonieri, Valeria Bonato, Beatrice Molena, Carlo Maestrone, Gianluca Airoldi, Carlo Olivieri, Pier Paolo Sainaghi, Federico Lionello, Giovanna Arcaro, Francesco Della Corte, Paolo Navalesi, Rosanna Vaschetto

**Affiliations:** 1Department of Cardiac Thoracic Vascular Sciences and Public Health, University of Padova, Via Gallucci, 13, 35121 Padova, Italy; gabriella.guarnieri@unipd.it (G.G.); beatrice.molena@unipd.it (B.M.); federico.lionello@aopd.veneto.it (F.L.); giovanna.arcaro@aopd.veneto.it (G.A.); 2Dipartimento di Medicina Traslazionale, Università del Piemonte Orientale, Via Solaroli, 17, 28100 Novara, Italy; nellodevita@hotmail.com (N.D.V.); lorenza.scotti@uniupo.it (L.S.); pierpaolo.sainaghi@med.unipmn.it (P.P.S.); francesco.dellacorte@med.uniupo.it (F.D.C.); rosanna.vaschetto@med.uniupo.it (R.V.); 3Pneumologia, Azienda Sanitaria Universitaria Giuliano Isontina, Via Giacomo Puccini, 50, 34148 Trieste, Italy; marco.confalonieri@asugi.sanita.fvg.it; 4Department of Anesthesia and Intensive Care, Azienda Ospedaliera SS. Antonio e Biagio e Cesare Arrigo, Via Venezia, 16, 15121 Alessandria, Italy; vbonato@ospedale.al.it; 5Anestesia Rianimazione ASL VCO, Dipartimento Chirurgico, Presidio Ospedaliero Domodossola e Verbania, Largo Caduti Lager Nazisti, 1, 28845 Domodossola, Italy; carlo.maestrone@aslvco.it; 6Medicina Interna, Ospedale Ss. Trinità, Viale Zoppis, 10, 28021 Borgomanero, Italy; gianluca.airoldi@asl.novara.it; 7Department of Anesthesia and Critical Care, Azienda Ospedaliera Sant’Andrea, Corso M. Abbiate, 21, 13100 Vercelli, Italy; carlo.olivieri@aslvc.piemonte.it; 8Istituto di Anestesia e Rianimazione, Dipartimento di Medicina-DIMED-Università di Padova, Azienda Ospedale-Università di Padova, Via Gallucci, 13, 35121 Padova, Italy; paolo.navalesi@unipd.it

**Keywords:** non-invasive respiratory support, octogenarian patients, COVID-19, acute respiratory failure

## Abstract

As the clinical outcome of octogenarian patients hospitalised for COVID-19 is very poor, here we assessed the clinical characteristics and outcomes of patients aged 80 year or older hospitalised for COVID-19 receiving non-invasive respiratory support (NIRS). A multicentre, retrospective, observational study was conducted in seven hospitals in Northern Italy. All patients aged ≥80 years with COVID-19 associated hypoxemic acute respiratory failure (hARF) undergoing NIRS between 24 February 2020, and 31 March 2021, were included. Out of 252 study participants, 156 (61.9%) and 163 (64.6%) died during hospital stay and within 90 days from hospital admission, respectively. In this case, 228 (90.5%) patients only received NIRS (NIRS group), while 24 (9.5%) were treated with invasive mechanical ventilation (IMV) after NIRS failure (NIRS+IMV group). In-hospital mortality did not significantly differ between NIRS and NIRS+IMV group (61.0% vs. 70.8%, respectively; *p* = 0.507), while survival probability at 90 days was significantly higher for NIRS compared to NIRS+IMV patients (0.379 vs. 0.147; *p =* 0.0025). The outcome of octogenarian patients with COVID-19 receiving NIRS is quite poor. Caution should be used when considering transition from NIRS to IMV after NIRS failure.

## 1. Introduction

Severe acute respiratory syndrome coronavirus 2 (SARS-CoV-2) infection, which causes coronavirus disease 2019 (COVID-19), was declared a global pandemic by the World Health Organization (WHO) on 11 March 2020. Disease severity from COVID-19 varies widely, and older adults are more likely to progress to severe disease, leading to high mortality rates [1]. In Italy, patients between 60–69, 80–89, and over 90 years old showed a case fatality rate (CFR) of 3.5%, 19.7%, and 22.7%, respectively [2,3]. During the first pandemic wave, in-hospital mortality rate in octogenarian COVID-19 patients in the New York City area was about 54% [4]. In a similar cohort of patients in Spain, the mortality rate was 33.3% [5]. Of interest, frailty was independently associated with lower survival in a population of 1346 patients with a median age of 75 years admitted to ICUs from 28 countries [6].

During the first pandemic wave, the use of mechanical ventilation (MV) for the management of COVID-19-associated hypoxemic acute respiratory failure (hARF) in older patients resulted in poor clinical outcomes. Indeed, a mortality rate of 72% was reported in 280 out of 388 subjects older than 80 years—68% of whom treated with invasive MV (IMV)—in German hospitals [7]. Adverse outcomes together with increasing concerns about ventilator shortages have led clinicians to consider other approaches to treat COVID-19 ARF patients, if medically appropriate, such as the use of non-invasive respiratory support (NIRS)—i.e., continuous positive airway pressure (CPAP) and/or non-invasive positive pressure ventilation (NPPV) [8]. To date, studies on COVID-19 focusing on clinical outcomes in older patients receiving NIRS due to hARF are scarce, showing a mortality rate ranging from 51.6% in octogenarian subjects receiving CPAP, NPPV, or high flow nasal cannula (HFNC) [9] to 70% in patients for whom NIRS is the ceiling of care [10,11,12].

Italy was one of the first European countries affected by COVID-19 and the evidence of age and frailty being associated with mortality was not initially known, particularly in those subjects receiving NIRS.

In our study, we aimed to describe the clinical characteristics and outcomes of patients aged 80 years or more hospitalized for COVID-19-associated hARF receiving NIRS at intermediate respiratory care units (IRCUs).

## 2. Methods

This is a multicentre, retrospective, cohort study based on medical records of three tertiary teaching hospitals (Padua, Novara and Trieste University Hospital) and four hospitals (Alessandria, Borgomanero, Domodossola and Vercelli hospital) in Northern Italy. All medical records of COVID-19 ARF patients aged ≥80 years treated with NIRS at IRCUs between 24 February 2020 and 31 March 2021 were collected and reviewed. The data were anonymised. Participating centres obtained ethics committee approval for the present research project (CE 243/21). Due to the retrospective nature of the study, the ethics committee waived the need for informed consent. This study was performed in accordance with the ethical standards of the 1964 Declaration of Helsinki and its later amendments or comparable ethical standards. Local investigators were responsible for ensuring data integrity and validity.

### 2.1. Participants

We retrospectively evaluated all patients aged ≥80 years diagnosed with severe pneumonia and laboratory confirmed SARS-CoV-2 infection [13] admitted to the IRCU of the participating hospitals for hARF who underwent CPAP or NPPV during the study period.

Hypoxemic ARF was defined as an acute, rapid deterioration in respiratory function and an exacerbation of dyspnoea over a few day time associated with a deterioration in blood gas tensions leading to hypoxemia—arterial oxygen tension to inspired oxygen fraction (PaO_2_/FiO_2_) ratio <250 mmHg during Venturi mask oxygen therapy [14]. Patients who received post-extubation CPAP/NPPV were excluded.

All hospital charts were reviewed for routinely collected patients’ demographic and clinical features, including age, gender, smoking habits, body mass index (BMI), and comorbidities, likely to influence outcome measures, such as chronic obstructive pulmonary disease (COPD), asthma, diabetes mellitus, cardiac disease—i.e., cardiac arrhythmia, previous myocardial infarction, angina pectoris, and/or congestive heart failure—and chronic renal failure. The Charlson comorbidity index score adjusted by age (ACCI) was also calculated for each patient [15].

On admission to the IRCU, the following parameters were recorded and analysed: time since onset of symptoms, Barthel index for activities of daily living [16] or Blaylock risk assessment screening score (BRASS) [17], respiratory rate (RR), heart rate (HR), body temperature, leukocyte count, D-dimer, serum C-reactive protein (CRP), arterial PaO_2_, PaCO_2_, and pH during spontaneous breathing with supplemental oxygen, and PaO_2_/FiO_2_. The sequential organ failure assessment (SOFA) score was also calculated [18]. According to the Barthel index score and BRASS, patients were considered at low (Barthel index > 80 and/or BRASS ≤ 10), moderate (Barthel index between 41 and 80 and/or BRASS between 11 and 19), or high (Barthel index ≤ 40 and/or BRASS ≥ 20) level of dependence, respectively [19].

### 2.2. Interventions

Details on CPAP/NPPV schedule, criteria for intubation, and IRCU organization are reported in the Appendix A. 

### 2.3. Outcome Measures and Statistical Analysis

Patients’ outcomes based on vital status parameters at both hospital discharge and 90-day follow-up after hospital admission, the length of stay at IRCU and hospital, and non-invasive, invasive, and total duration of MV, were analysed.

Descriptive statistics were performed to summarise the patients’ characteristics. Categorical variables were reported as absolute frequencies and percentages, whilst continuous variables were reported as median and first and third quartiles, as they were not found to be normally distributed. The Shapiro-Wilks test was used to assess whether or not continuous variables followed the normal distribution. Mann-Whitney and Chi square tests were used to assess the differences in patients’ characteristics between the NIRS and NIRS+IMV group.

Univariable Cox proportional hazard models were used to calculate the hazard ratios (HRs) in order to evaluate a possible association between the patients’ characteristics and the risk of death at 90 days. The Simon and Makuch’s method was applied to draw survival curves stratified by treatment (NIRS vs. NIRS+IMV), accounting for the time-dependent nature of the stratification variable. The Mantel and Bayar test was used to assess the differences in survival curves Univariable and multivariable Cox proportional hazard models were used to analyse the relationship between treatment and 90-day mortality adjusted for selected confounders. A stepwise procedure was used to select the variables to be included in the multivariable model. As for survival curves, treatment was included in the model as time-dependent variable to avoid immortal time bias. All statistical analyses were conducted using SAS software for Windows, version 9.4 (SAS Institute, Cary, NC, USA).

## 3. Results

### 3.1. Patient Characteristics

During the study period, 252 patients aged ≥80 years admitted to IRCUs for hARF with a primary diagnosis of COVID-19 severe pneumonia were prescribed NIRS, administered as CPAP/NPPV, and were included in our retrospective study. Among them, 228 (90.5%) received CPAP/NPPV as the highest level of respiratory assistance (NIRS group), while 24 (9.5%) received IMV via endotracheal intubation (ETI) after CPAP/NPPV failure (NIRS+IMV group).

The patients’ baseline demographic and clinical characteristics and clinical and laboratory data on IRCU admission are outlined in Table 1. When compared with NIRS+IMV patients, NIRS patients were on average two years older (84 vs. 82 years; *p* = 0.0183), had higher PaCO_2_ (34 vs. 32 mmHg, *p* = 0.0449) and lower serum CRP (11 vs. 15 mg/dL; *p =* 0.0280) levels, and displayed a lower SOFA score (3 vs. 5; *p =* 0.0338).

### 3.2. Clinical Outcomes

Out of 252 study participants, 156 (61.9%) and 163 (64.6%) died during hospital stay and within 90 days from hospital admission, respectively. Among 228 patients in the NIRS group, 139 (61.0%) died during hospital stay, while 145 (63.6%) were deceased at 90-day follow-up. Out of 24 patients in the NIRS+IMV group, 17 (70.8%) and 18 (75%) died during hospital stay and within 90 days from hospitalization, respectively. The proportion of deaths during hospitalization and at 90-day follow-up were not significantly different between NIRS and NIRS+IMV group (*p* = 0.507 and *p* = 0.3697, respectively). Data on patient clinical course are illustrated in Figure 1.

Table 2 outlines the HRs for the association between the patients’ characteristics and the risk of 90-day mortality according to univariable Cox models. Older age, increasing CRP, SOFA score, and level of dependence were all associated with a statistically significant increased risk of death at 90 days, whilst higher levels of SaO_2_ and longer duration of MV were associated with a decreased risk of death.

Figure 2 shows Simon and Makuch’s curves stratified by treatment. The median survival time was 22 (9.0-NA) days for patients in the NIRS group and 12 (8.0–21.0) days for those in the NIRS+IMV group.. Furthermore, the survival probability at 90 days of NIRS and NIRS+IMV patients was 0.379 (risk of death: 0.621) and 0.147 (risk of death: 0.835), respectively. According to the Mantel and Bayar’s test, the curves were significantly different (*p*
*=* 0.0025). Although NIRS+IMV patients showed a risk of dying at 90-day follow-up 2.18 times higher (95%CI 1.33–3.56) than that observed in patients only treated with NIRS, this increased risk failed to reach a statistically significant level after adjustment for age, ACCI, CRP, SOFA score, and SaO_2_ (Table 3).

The total duration of MV was significantly longer in the NIRS+IMV group compared to the NIRS group (8 vs. 6 days, *p =* 0.0121). The time spent on NIRS was similar between the two groups, with a median time of 5 days; the median duration of IMV in the NIRS+IMV group was 4 (0–7) days.

## 4. Discussion

The present study shows that both in-hospital and 90-day follow-up mortality rates were above 60% (61.9% and 64.6%, respectively) in a large cohort of octogenarians receiving NIRS for COVID-19 hARF. Moreover, patients undergoing IMV after NIRS failure, representing 9.5% of our study population, had a significantly lower survival probability at 90 days than that of patients receiving NIRS as a ceiling of care. After adjusting mortality for age, ACCI, CRP, SOFA score, and SaO_2_, no additional benefit in terms of survival was found for patients in the NIRS+IMV group.

Recently, a number of studies have focussed on the mortality risk of older patients hospitalised for COVID-19-related ARF, showing that 40% of octogenarians admitted to hospital for COVID-19 pneumonia did not survive at 28-day from hospital admission [20], and that the mortality rate increased up to 50% among patients receiving out-of-ICU NIRS [8]. Moreover, case fatality rate of patients aged 80 years or older admitted to ICU for COVID-19-associated ARDS was as high as 84% [21], which is a much higher value than that of same age patients with non-COVID-19-related ARDS requiring planned or unplanned ICU admission [22,23]. Furthermore, Lim et al., reported that COVID-19 octogenarian patients undergoing IMV have a higher mortality rate compared to patients of similar age admitted to ICU and treated with NIRS [21].

An association between increasing age and higher NIRS failure rate was reported by several observational studies [24,25,26]. In particular, a prospective cohort study on patients undergoing NIRS due to de novo hypoxemic ARF showed that patients aged ≥80 years had much higher ICU and hospital mortality rates than those of patients aged <80 years (28% vs. 17%, *p* = 0.03, and 40% vs. 25%, *p* < 0.01, respectively), with the highest values recorded in the ‘do-not-intubate’ (DNI) subgroup [27]. This led the authors to recommend caution when considering NIRS in very old patients. In contrast, a subsequent retrospective, observational study on patients undergoing NIRS to treat ARF of various aetiologies (i.e., COPD exacerbation, acute pulmonary oedema, sepsis, and pneumonia) [28] showed that the in-hospital mortality rate of very old individuals (≥85 years) was comparable to that observed in younger patients, despite a higher predicted mortality in the former population. We believe that these contrasting observations may be attributable to the fact that NIRS outcomes in old age patients can be heavily influenced by the baseline disease, with hypoxemic ARF being the least responsive to treatment [28].

Given the high NIRS failure rate and the overall poor survival rate in patients aged ≥80 years, the identification and implementation of appropriate selection criteria for potential NIRS candidates would allow the development of a proper strategy for allocation of IRCU resources during a pandemic. In this regard, when patient characteristics associated with NIRS failure were analysed, we found that increased CRP, SOFA, and ACCI values were associated with a statistically significant higher risk of death at 90 days. Whilst these parameters have been reported to be sensitive prognostic indicators of in-hospital mortality in a population of COVID-19 patients distributed over a wide age range [11,29,30,31], to our knowledge this is the first study showing their importance in predicting a successful outcome of NIRS among patients aged 80 year or older. Moreover, in our patients the level of dependence on IRCU admission appears to be significantly associated with the risk of mortality. Indeed, patients with increased levels of frailty (Barthel index ≤ 40 and/or BRASS ≥ 20) had a substantially higher risk of mortality (over 2-fold) compared to that of patients with a low and/or moderate dependence level. In good agreement, a retrospective study on a group of 100 very old COVID-19 cases (median age: 85 years) in a long-term care setting revealed a strong correlation between Barthel index and mortality [32].

The decision to offer IMV to very old patients responding poorly to NIRS is still largely debated. In our study, 9.5% patients were switched to IMV after experiencing NIRS failure: they were found to be slightly younger, had a significantly higher SOFA score, and showed slightly higher CRP values compared to patients who were not intubated. Of note, the clinical outcomes of patients undergoing IMV were basically the same as those of the NIRS group. Our findings are in line with those reported by a prospective cohort study of Schortgen et al., that examined 163 very old patients admitted to Intensive Care Unit (ICU) who received non-invasive ventilation (NIV) for standard indications (i.e., chronic obstructive pulmonary disease (COPD) exacerbation and/or cardiogenic pulmonary oedema), demonstrating that the 6-month mortality rate was 67% for DNI patients increasing to 77% in patients switched to IMV following NIRS failure [27]. Indeed, Lerolle et al., claimed that complications occurring during IMV in older patients (≥80 years), reflecting frailty of old age and multiple comorbidities, may explain why the risk of hospital mortality is increased in very old individuals undergoing invasive ventilatory treatment [33].

Being retrospective and observational, our study has some limitations (e.g., unknown confounding factors, selection bias etc.), which would have been avoided by performing a prospective analysis. It should be however considered that prospective randomized trials are unlikely to be conducted in a pandemic scenario such as the one we have been experiencing in the last two years. In addition, because of our study design, we must admit that our findings may not be extended to other conditions due to the exceptionality of the COVID-19 pandemic outbreak. Furthermore, as our study is based on NIRS techniques implemented at IRCUs, our results cannot be applied to same age patients managed in the ward. Finally, the decision of excluding DNI patients from intubation and therapeutic up-grade was not based on standardized protocols but rather on clinical judgment.

## 5. Conclusions

The clinical outcome of patients aged 80 years or older with COVID-19-associated ARF receiving NIRS at IRCUs is very poor, and increased levels of frailty on IRCU admission are associated with an even poorer prognosis. Given the extremely poor prognosis of patients intubated after NIRS failure, caution should be used when considering seriously ill older adults for IMV after unsuccessful NIRS.

Altogether, our results may favour a more efficient use of IRCU resources during future COVID-19 outbreaks, particularly in settings where the availability of clinicians, beds and equipment is limited.

## Figures and Tables

**Figure 1 jcm-11-01372-f001:**
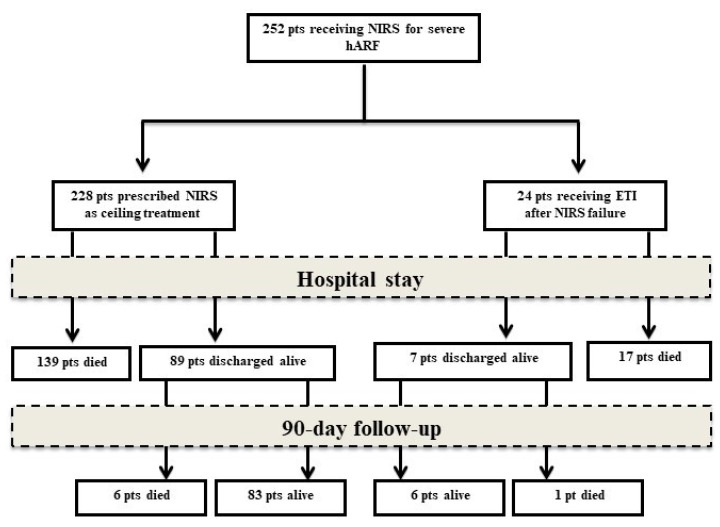
Study profile. NIRS, non-invasive respiratory support; ETI, endotracheal intubation.

**Figure 2 jcm-11-01372-f002:**
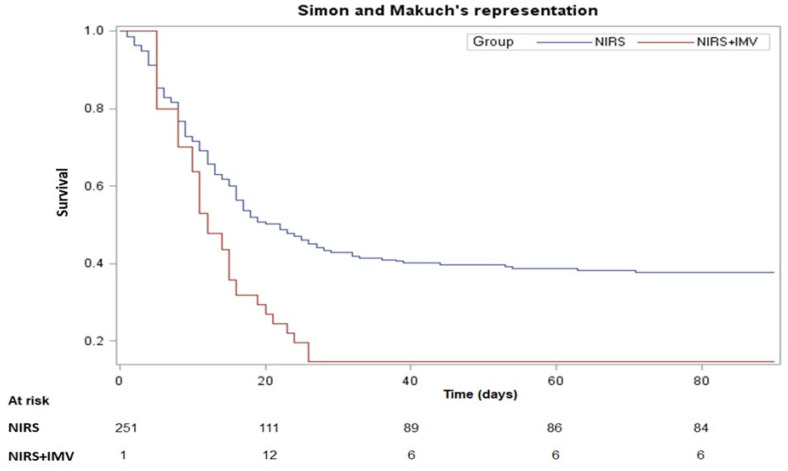
Simon and Makuch’s survival curves stratified by treatment.

**Table 1 jcm-11-01372-t001:** Patients’ baseline demographic and clinical characteristics, clinical and laboratory data on Intermediate Respiratory Care Unit admission, and clinical outcomes.

	Overall (n = 252)	NIRS (n = 228)	NIRS+IMV (n = 24)	*p*-Value
**Baseline demographic and clinical data**				
Age, years	84 (82–87)	84 (82–87)	82 (81–84)	0.0183
Female, n	92 (37)	86 (38)	6 (25)	0.2183
Non-smokers, n	158 (73)	148 (73)	10 (63)	0.3867 ^
Level of dependence				0.3167
-low	74 (42)	68 (42)	6 (35)	
-moderate	27 (15)	26 (16)	1 (6)	
-high	77 (43)	67 (42)	10 (59)	
Body mass index, kg/m^2^	26 (23–29)	26 (23–29)	24 (22–29)	0.1536
ACCI	6 (4–7)	6 (4–7)	6 (4–8)	0.7976
**Clinical, laboratory and blood gas data on IRCU admission**				
Time since symptom onset, days	5 (3–8)	5 (3–8)	6 (3–10)	0.6606
Heart rate, beats/min	83 (74–95)	82 (74–95)	85 (75–94)	0.7098
Respiratory rate, breaths/min	26 (22–30)	26 (22–30)	25 (18–27)	0.0726
Temperature, Celsius	36.5 (36–37)	36.5 (36–37)	36.8 (36.7–37)	0.1179
White blood cell count, ×10^3^/μL	8.5 (6.1–12.1)	8.5 (6.1–12.1)	8.6 (6.9–12.1)	0.5512
D-dimer, μgFEU/L	628 (238–1900)	620 (241–1691)	1317 (207–3180)	0.2915
Serum C-reactive protein, mg/dL	11 (6–16)	11 (6–15)	15 (10–25)	0.0280
PaO_2_, mmHg	56 (45–69)	56 (46–69)	53 (41–73)	0.4886
PaCO_2_ mmHg	34 (30–38)	34 (31–38)	32 (28–37)	0.0449
Arterial pH	7.45 (7.42–7.49)	7.45 (7.42–7.48)	7.46 (7.42–7.49)	0.5132
SaO_2_, %	89 (82–93)	89 (83–93)	86 (76–95)	0.3328
PaO_2_/FiO_2_, mmHg	110 (80–172)	111 (81–171)	100 (61–190)	0.4361
SOFA score	4 (3–5)	3 (3–4)	5 (3–6)	0.0338
**Clinical outcomes**				
Days on mechanical ventilation	6 (3–10)	6 (2–10)	8 (6–13)	0.0121
Days on non-invasive ventilation	5 (2–10)	5 (2–10)	5 (1–9)	0.1870
Days on invasive ventilation *	NA	NA	4 (0–7)	NA
Length of IRCU stay, days	8 (4–13)	8 (4–13)	5 (1–11)	0.1126
Length of hospital stay, days	14 (8–23)	14 (8–23)	18 (11–26)	0.1968
Pts discharged alive, n (%)	96 (38)	89 (39)	7 (29)	0.7636
Pts alive at 90-day follow-up, n (%)	89 (35)	83 (36)	6 (25)	0.3854

Data are presented as median (interquartile range) or number (percentage), *P*-values refer to differences between “NIRS” and “NIRS+IMV”. NIRS, non-invasive respiratory support; IMV, invasive mechanical ventilation; ACCI, age adjusted Charlson comorbidity index; PaO_2_/FiO_2_, arterial partial pressure of oxygen to inspired oxygen fraction ratio; SaO_2_, arterial oxygen saturation; SOFA, sequential organ failure assessment; NA, not applicable; IRCU, intermediate respiratory care unit. ^ Fisher exact test. * Calculated on patients undergoing endotracheal intubation.

**Table 2 jcm-11-01372-t002:** Distribution of patient characteristics according to death status at 90 days after hospitalization and hazard ratio (HR) and 95% confidence interval (95%CI) derived from the univariable Cox proportional hazard model.

	Deceased at 90-Day Follow-Up			
	No (n = 89)	Yes (n = 163)	All (n = 252)	HR (95%CI)
Age, years	83 (81–85)	85 (82–87)	84 (82–87)	1.064 (1.022–1.107)
Female (M/F), %	33 (37)	59 (36)	92 (37)	1.097 (0.797–1.510)
Non-smokers, n	58 (70)	100 (74)	158 (73)	1
Level of dependence, n				
-low	39 (57)	35 (32)	74 (42)	1
-moderate	14 (21)	13 (12)	27 (15)	1.020 (0.540–1.927)
-high	15 (22)	62 (56)	77 (43)	2.217 (1.462–3.362)
Body mass index, kg/m^2^	26 (23–29)	26 (24–29)	26 (23–29)	0.980 (0.930–1.033)
ACCI	5 (4–7)	6 (4–7)	6 (4–7)	1.053 (0.995–1.114)
Time since symptom onset, days	5 (3–10)	5 (3–8)	5 (3–8)	0.979 (0.945–1.014)
Heart rate, beats/min	80 (74–94)	83 (74–98)	83 (74–95)	1.008 (0.998–1.018)
Respiratory rate, breaths/min	26 (22–30)	26 (22–30)	26 (22–30)	1.015 (0.988–1.043)
Temperature, Celsius	36.5 (36–37)	36.5 (36–37)	36.5 (36–37)	1.097 (0.833–1.446)
White blood cell count, ×10^3^/μL	8.24 (6.13–11)	8.79 (6.17–12.64)	8.5 (6.13–12.1)	1.006 (0.979–1.035)
D-dimer, μgFEU/L	524 (198–1544)	711 (241–2356)	628 (238–1900)	1.000 (1.000–1.000)
Serum C-reactive protein, mg/dL	9.59 (5.3–14.76)	11.44 (7.01–16.9)	11 (6.36–15.9)	1.037 (1.016–1.059)
PaO_2_, mmHg	59 (47–75)	53 (44–67)	56 (45–69)	0.992 (0.984–1.000)
PaCO_2_ mmHg	35 (31–38)	33 (30–37)	34 (30–38)	0.992 (0.973–1.011)
Arterial pH	7.46 (7.42–7.49)	7.45 (7.41–7.48)	7.45 (7.42–7.49)	0.463 (0.053–4.045)
SaO_2_, %	90 (84–94)	88 (82–93)	89 (82–93)	0.975 (0.959–0.992)
PaO_2_/FiO_2_, mmHg	120 (88–179)	104 (76–170)	110 (79–172)	0.998 (0.995–1.001)
SOFA score	3 (2–4)	4 (3–5)	4 (3–5)	1.195 (1.111–1.285)
Days on mechanical ventilation	7 (3–12)	5 (2–8.5)	6 (3–10)	0.946 (0.919–0.974)

ACCI, age-adjusted Charlson comorbidity index; HR, hazard ratio; PaO_2_, arterial partial pressure of oxygen; PaCO_2_, arterial partial pressure of carbon dioxide; PaO_2_/FiO_2_, arterial partial pressure of oxygen to inspired oxygen fraction ratio; SaO_2_, arterial oxygen saturation; SOFA, sequential organ failure assessment.

**Table 3 jcm-11-01372-t003:** Hazard ratios (HRs) for the association between ventilatory treatment and the risk of 90-day mortality according to univariable and multivariable Cox proportional hazard models.

	Univariable Model HR (95% CI)	Multivariable Model HR (95% CI)
NIRS+IMV vs. NIRS	2.17 (1.33–3.56)	1.76 (0.86–3.60)
Age		1.11 (1.05–1.17)
ACCI		1.11 (1.03–1.18)
CRP		1.05 (1.02–1.07)
SOFA score		1.16 (1.06–1.26)
SaO_2_		0.97 (0.95–0.99)

NIRS, non-invasive respiratory support; IMV, invasive mechanical ventilation; ACCI, age-adjusted Charlson comorbidity index; CRP, C-reactive protein; SOFA, sequential organ failure assessment; SaO_2_, arterial oxygen saturation; HR, hazard ratio; CI, confidence interval.

## Data Availability

The data that support the findings of this study are available from the corresponding author, [A.V.], upon reasonable request.

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
