# Peer review of "Clinical Outcomes in Patients Aged 80 Years or Older Receiving Non-Invasive Respiratory Support for Hypoxemic Acute Respiratory Failure Consequent to COVID-19"

_jcm, 2022, doi:10.3390/jcm11051372_

Round 1
Reviewer 1 Report
The article is very interesting and very practical. It has the character of a register, concentrated on survival of the patients. However, the results of survival in each of the groups must be extracted from the text. It is accepted to present the results in this type of work in the form of a clear scheme. There is also a lack of transparent information about the time of stay in the hospital, about the number of patients discharged.
Conclusions from the work are of significant practical importance. This article may facilitate the decision not to qualify some patients to respiratorotherapy due to its proven ineffectiveness (in this group of patients and in this disease). This article can help in a significant reduction of the effort of nurses and doctors in the burden on insufficient medical Staff, specialist departments and equipment.
Author Response
We thank the reviewer for the suggestions provided and the helpful comments.
The article is very interesting and very practical. It has the character of a register, concentrated on survival of the patients. However, the results of survival in each of the groups must be extracted from the text.
The number of patients alive at discharge and 90-day follow-up has been outlined in Table 1.
It is accepted to present the results in this type of work in the form of a clear scheme.
Agreed. Figure 1 has been added, illustrating the study profile.
There is also a lack of transparent information about the time of stay in the hospital, about the number of patients discharged.
Length of hospital stay was already outlined in Table 1. The number of patients discharged alive has been added in Table 1.
Reviewer 2 Report
Dear authors, thank you for this article- i think it is well researched, well written and presents findings that were clearly appropriate for the alpha/delta variants which i think were predominant in your patients. However, your findings are not original and are well known. Age and frailty are associated with increased mortality in the context of ARDS and COVID 19 and there have been a large number of reviews published on this. Composite scores such as the ISARIC 4C scores have been created to risk stratify patients. I would quite like to know how were patients chosen to have non-invasive ventilation or CPAP? Surely those with high level of dependence should not have been offered a measure which would have proved futile and thus palliative care would have proven more useful?
Perhaps I am being harsh, and I know that Italy was one of the first European countries affected, and thus the evidence of frailty being associated with mortality was perhaps not initially known and perhaps more emphasis should be placed on this.
I would review again. Please look up the COVIP studies as well as Age and frailty are independently associated with increased COVID-19 mortality and increased care needs in survivors: results of an international multi-centre study | Age and Ageing | Oxford Academic (oup.com)
Author Response
We thank the reviewer for the suggestions provided and the helpful comments.
Dear authors, thank you for this article- i think it is well researched, well written and presents findings that were clearly appropriate for the alpha/delta variants which i think were predominant in your patients. However, your findings are not original and are well known. Age and frailty are associated with increased mortality in the context of ARDS and COVID 19 and there have been a large number of reviews published on this. Composite scores such as the ISARIC 4C scores have been created to risk stratify patients.
The reviewer is clearly right when suggesting that an association between age and poor outcomes in individuals with COVID-19 has been already demonstrated. Worth remarking, however, studies on COVID-19 focusing on older patients receiving NIRS are very scarce. Moreover, our study makes clear for the first time that transition from non-invasive to invasive ventilation can be futile in older patients.
I would quite like to know how were patients chosen to have non-invasive ventilation or CPAP?
A treatment algorithm based on a stepwise utilization of High Flow Nasal Oxygen, Continuous Positive Airway Pressure (CPAP) or Non-Invasive Ventilation (NIV), and endotracheal intubation (ETI) was utilized by most participating Intermediate respiratory care units in the effort to reverse hypoxemia in patients with COVID-191. In this context, the decision to proceed to CPAP or NIV was taken by the attending physician depending on local training habits.
1 Pasin L, Sella N, Correale C, et al. Regional COVID-19 Network for Coordination of SARS-CoV-2 outbreak in Veneto, Italy. J Cardiothorac Vasc Anesth. 2020;34(9):2341-5.
Surely those with high level of dependence should not have been offered a measure which would have proved futile and thus palliative care would have proven more useful?
As pointed out by the reviewer, Italy was one of the first European countries affected by the pandemic. In such a situation, lacking of sufficient knowledge on patients’ outcomes made it very difficult to predict whether NIRS application could be futile and palliative care should be preferred.
Perhaps I am being harsh, and I know that Italy was one of the first European countries affected, and thus the evidence of frailty being associated with mortality was perhaps not initially known and perhaps more emphasis should be placed on this.
Agreed. A short paragraph emphasizing that Italy was the first European nation to be affected by COVID-19 has been added in the “Introduction” section
Please look up the COVIP studies as well as Age and frailty are independently associated with increased COVID-19 mortality and increased care needs in survivors: results of an international multi-centre study | Age and Ageing | Oxford Academic (oup.com)
Agreed. The results obtained from the COVIP study have been added in the “Introduction” section.
Round 2
Reviewer 2 Report
I am happy with the changes, and this can now be published